# Yeast Nup84-Nup133 complex structure details flexibility and reveals conservation of the membrane anchoring ALPS motif

Sarah A. Nordeen [1], Daniel L. Turman[1] & Thomas U. Schwartz [1✉]

The hallmark of the eukaryotic cell is the complex endomembrane system that compartmentalizes cellular functions. Transport into and out of the nucleus occurs through the nuclear pore complex (NPC). The heptameric Nup84 or Y complex is an essential scaffolding component of the NPC. Here we report two nanobody-bound structures: the full-length Nup84-Nup133 C-terminal domain complex and the Nup133 N-terminal domain, both from *S. cerevisiae*. Together with previously published structures, this work enables the structural description of the entire 575 kDa Y complex from one species. The structure of Nup84-Nup133$_{CTD}$ details the high flexibility of this dimeric unit of the Y complex. Further, the Nup133$_{NTD}$ contains a structurally conserved amphipathic lipid packing sensor motif, confirmed by liposome interaction studies. The presented structures reveal important details about the function of the Y complex that affect our understanding of NPC structure and assembly.

[1] Department of Biology, Massachusetts Institute of Technology, Cambridge, MA, USA. ✉email: tus@mit.edu

In the eukaryotic cell, the genetic material is sequestered in the nucleus, separated from the cytoplasm by the double-membraned nuclear envelope (NE). Transport of soluble cargos across the NE occurs solely through nuclear pore complexes (NPCs), massive, proteinaceous assemblies that sit in circular openings where inner and outer nuclear membranes fuse, effectively perforating the NE. Regulated transport through NPCs plays a key role in many essential cellular processes, such as regulated gene expression, cell division, and ribosome assembly[1]. The NPC is a cylindrical assembly subdivided into three rings, based on cryo-electron tomographic analysis[2–4]. The cytoplasmic and nucleoplasmic rings each sit on the surface of the NE, sandwiching the inner ring in between. Despite the NPC's large mass (~60MDa in yeast), only about 30 different nucleoporins (nups) make up the 8-fold symmetric assembly of subcomplexes that together form the NPC[5–9]. These Nups can be classified into three main categories, (i) scaffold Nups that make up the stable structure of the NPC, (ii) peripheral Nups that are biased to one face of the NPC, and (iii) phenylalanine-glycine (FG) Nups that form the permeability barrier of the NPC.

One main scaffold Nup subcomplex is the Nup84 or Y complex[5]. The Y complex is an essential component for NPC assembly and integrity, and it makes up ~20% of the mass of the NPC[10,11]. In *Saccharomyces cerevisiae*, the heptameric 575 kDa Y complex comprises Nup84, Nup85, Nup120, Nup133, Nup145C, Sec13, and Seh1[12]. It can generally be described as having two short arms and a long stalk, connected in a central hub. Nup120 and the Seh1-Nup85 complex form one short arm each, while Nup145C-Sec13, Nup84, and Nup133 sequentially emanate from the hub to form the long stalk (Fig. 1a). Great efforts to study the Y complex via X-ray crystallography yielded structures of individual Y complex Nups or overlapping subassemblies across multiple species to near completion[13–27]. However, there remains one elusive element of the Y complex never solved by X-ray crystallography from any species: the full-length structure of Nup84. While complex structures of an N-terminal fragment with

Nup145C-Sec13, and a C-terminal fragment of Nup107 (the human homolog) with Nup133 are known (Fig. 1b, c), the missing connection between the two yields a significant problem in modeling a composite Y complex structure. Several studies suggested flexibility within the middle segment of Nup84/Nup107, but the degree of this and whether the modeling of the missing element is correct are limiting uncertainties[26–29]. Nup84 has an ancestral coatomer element (ACE1) fold, as do Nup85, Nup145C, and Nic96 of the inner ring complex[6]. The ACE1 fold is characterized by three elements, the crown, the trunk, and the tail, formed by a specific helical arrangement that forms a fold-back, U-shaped structure[14]. Flexibility has been observed at the interfaces between the three core elements[30]. However, for Nup84, the similarity with other scaffold nups suggests that it should not contain a significantly disordered region. Therefore, we reasoned that knowing the structure of full-length Nup84 would be critical to better model the entire Y complex. The structure of the assembled NPC, which remains a major goal in structural biology, can only be revealed by a hybrid approach, combining high-resolution component structures with lower-resolution cryo-ET maps capturing the NPC in situ[2,4,31–33].

In addition to the lack of knowledge about Nup84, we have little structural information about yeast Nup133 aside from the C-terminal heel domain, which only has modest similarity to its human homolog[23,24] (Fig. 1b). Also, completing the structure of the Y complex from a single species should help understand the differences in the NPC structure across species[33–35]. As a member of the cytoplasmic and nucleoplasmic rings in the NPC, the Y complex coats the peripheral inner and outer nuclear membrane (INM, ONM, respectively) bordering the circular openings in the NE[36]. It remains unclear how the yeast Y complex interacts with the INM and ONM and how it links to the inner NPC ring. The ArfGAP1 lipid packing sensor (ALPS) motif in the human homolog of Nup133 is thought to anchor the Y complex to the INM and ONM[22,37,38]. The hsNup133 ALPS motif is critical for interphase assembly in metazoa, a process which is thought to be

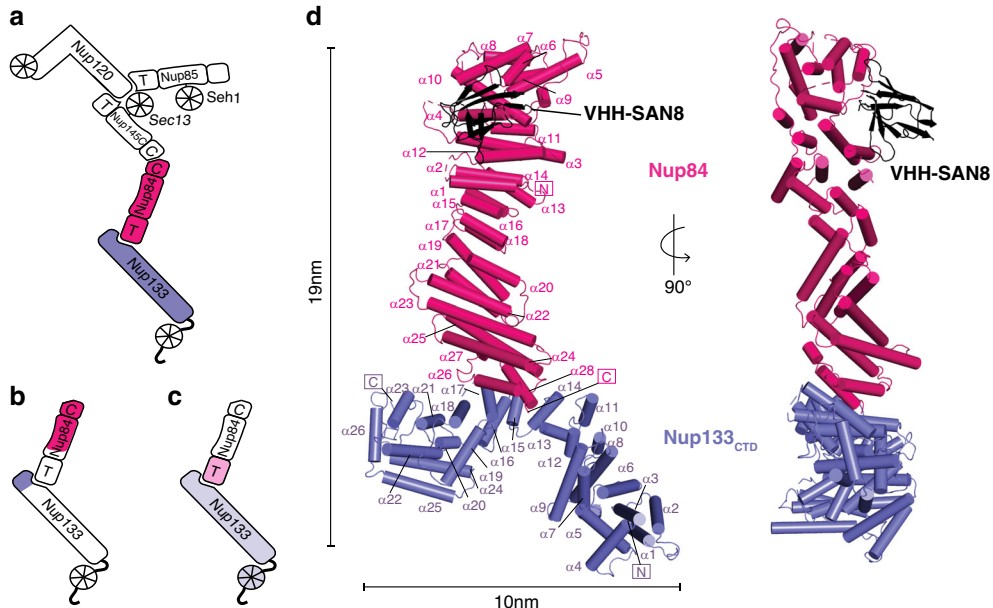

**Fig. 1 Structure of the *Saccharomyces cerevisiae* Nup84-Nup133_CTD-VHH-SAN8 complex. a** Schematic of the *S. cerevisiae* Y complex. Regions included in the structure are colored (Pink—Nup84, Purple—Nup133). Elements of ACE1 fold proteins are indicated: T—tail and C—crown adjacent to the central trunk element. **b** Schematic of *S. cerevisiae* Nup84-Nup133 with previously solved fragments colored as in **a**. **c** Schematic of homologous human Nup107-Nup133 with previously solved fragments colored in light pink—Nup107 and light purple—Nup133. **d** Structure of Nup84-Nup133_CTD-VHH-SAN8, with Nup84 in pink, Nup133_CTD in purple, and VHH-SAN8 in black. N and C termini are indicated and helices are numbered. Right panel shows the side view (90° rotation) of the complex.

**Table 1 Data collection and refinement statistics.**

| Protein | Nup84-Nup133$_{521-1157}$ VHH-SAN8 | Nup84-Nup133$_{521-1157}$ VHH-SAN8 TeW derivative | Nup84-Nup133$_{521-1157}$ | Nup84-Nup133$_{521-1157}$ VHH-SAN8/9 | Nup133$_{55-481}$ VHH-SAN4 | Nup133$_{55-481}$ VHH-SAN5 |
|---|---|---|---|---|---|---|
| PDB code | 6X02 | – | – | 6X03 | 6X05 | 6X04 |
| Organism | | *S. cerevisiae, V. pacos* | | | | |
| Data collection | | | | | | |
| Space group | P2$_1$2$_1$2$_1$ | P2$_1$2$_1$2$_1$ | P2$_1$2$_1$2$_1$ | P2$_1$2$_1$2$_1$ | P2$_1$2$_1$2 | P2$_1$2$_1$2$_1$ |
| Cell dimensions | | | | | | |
| a, b, c (Å) | 72.3, 287.7, 297.4 | 72.5, 283.6, 295.7 | 73.7, 298.3, 310.0 | 72.3, 295.2, 295.6 | 93.4, 154.7, 41.4 | 85.7, 204.7, 205.2 |
| α, β, γ (°) | 90, 90, 90 | 90, 90, 90 | 90, 90, 90 | 90, 90, 90 | 90, 90, 90 | 90, 90, 90 |
| Resolution (Å) | 103.4-6.4 (6.6-6.4)$^a$ | 147.8-7.8 (8.0-7.8) | 147.8-8.3 (8.6-8.3) | 147.8-7.4 (7.6-7.4) | 59.6-2.1 (2.2-2.1) | 49.7-2.7 (2.8-2.7) |
| R$_{p.i.m.}$ | 2.2 (81.3) | 4.2 (42.0) | 3.9 (39.4) | 2.7 (61.7) | 5.2 (48.0) | 3.5 (68.4) |
| I/σ | 55.3 (1.0) | 13.8 (2.1) | 27.0 (1.8) | 46.6 (1.1) | 19.3 (1.5) | 29.1 (1.5) |
| CC$_{1/2}$ | 1.00 (0.58) | 1.00 (0.89) | 0.99 (0.79) | 0.97 (0.54) | 1.00 (0.51) | 1.00 (0.57) |
| Completeness (%) | 99.8 (100.0) | 99.9 (100.0) | 99.8 (99.5) | 99.9 (100.0) | 99.9 (99.8) | 99.6 (97.3) |
| Redundancy | 24.1 (25.5) | 12.3 (12.9) | 6.8 (7.2) | 12.0 (12.7) | 11.9 (10.3) | 8.6 (8.3) |
| Refinement | | | | | | |
| Resolution (Å) | 103.4-6.4 | | | 147.8-7.4 | 59.6-2.1 | 49.7-2.7 |
| No. of Reflections | 13,453 | | | 9,375 | 35,967 | 100,268 |
| R$_{work}$/R$_{Free}$ | 31.4/32.4 | | | 33.5/35.3 | 18.3/22.3 | 23.6/25.9 |
| No. of atoms | | | | | | |
| Protein | 6655 | | | 7106 | 3890 | 20,184 |
| Water | 0 | | | 0 | 285 | 17 |
| B factors (Å$^2$) | | | | | | |
| Protein | 574.6 | | | 668.3 | 43.0 | 88.5 |
| Water | — | | | — | 43.0 | 46.9 |
| r.m.s. deviations | | | | | | |
| Bond length (Å) | 0.003 | | | 0.003 | 0.007 | 0.003 |
| Bond angles (°) | 0.84 | | | 0.81 | 0.87 | 0.57 |

$^a$Values in parenthesis are for highest-resolution shell (10% of the data). One crystal was used for each dataset.

similar to NPC assembly in organisms with closed mitosis like *S. cerevisiae*[38,39]. Whether or not the ALPS motif is conserved in *S. cerevisiae* has been debated in the literature, due to the lack of high-resolution structural or functional studies on yeast Nup133[22,25,37].

Here we report the structures of the Nup84-Nup133 C-terminal α-helical domain and Nup133 N-terminal β-propeller *from S. cerevisiae*. The structures were obtained using nanobodies, single domain antibodies derived from alpacas[40], as crystallization chaperones. This completes the entire structure of the Y complex from *S. cerevisiae*, allowing us to create a complete composite model of the Y complex assembly from a single species. Additionally, we show that Nup133 has a functional ALPS motif through liposome interaction studies. This model of the Y complex delineates key hinge points and possible motion ranges in the Y complex stalk and establishes the position of Nup133 with its ALPS motif placed adjacent to the membrane in the NPC assembly.

## Results

**Structure of the Nup84-Nup133CTD-VHH-SAN8 complex.** We solved the structure of full-length *S. cerevisiae* Nup84-Nup133 C-terminal domain (Nup133$_{CTD}$, residues 521–1157) bound by a nanobody (VHH-SAN8) by single-wavelength anomalous dispersion (SAD) to 6.4 Å resolution using an Anderson-Evans polyoxotungstate derivative (Table 1, Fig. 1d). We observed clear, elongated, helical density in the initial, solvent-flattened experimental map. After multiple rounds of refinement and successively placing helices, we arrived at a map

with clear enough density to employ both real-space docking and molecular replacement to place fragments of the structure that were previously known[18,23,24] (Supplementary Fig. 1). Following placement of roughly half of Nup84 and the C-terminal heel domain of Nup133, we were able to manually build the remaining helices. Due to the limited resolution of the data, we did not build side chains for the model. However, we are confident in assigning the sequence for both, Nup84 and Nup133 (see Material and Methods for details).

Placement of the nanobody VHH-SAN8 was unambiguous as we observed large difference density near the Nup84 crown element (Supplementary Fig. 2). However, the electron density around the three complementarity determining regions (CDRs), which presumably make up the majority of the interface with Nup84 were not clear enough for chain tracing. Therefore, we only rigid body positioned a nanobody model without CDRs in our structure. Additionally, we solved the structure of Nup84-Nup133$_{CTD}$ bound to both VHH-SAN8 and a second nanobody, VHH-SAN9, at 7.4 Å resolution (Table 1 and Supplementary Fig. 3). The doubly bound Nup84-Nup133$_{CTD}$-VHH8/9 structure crystallized in the same crystal form as the singly bound Nup84-Nup133$_{CTD}$-VHH8 structure. VHH-SAN9 binds on the opposite face as VHH-SAN8 at nearly the same vertical position along Nup84. Again, this resolution precluded the building of the CDR loops for both nanobodies, and a nanobody model without CDRs was rigid body positioned into the map at the site of clear difference density for VHH-SAN9 (Supplementary Fig. 2). We speculate that the limited resolution of both structures is due to crystals with very high solvent content (~85%) and rather small

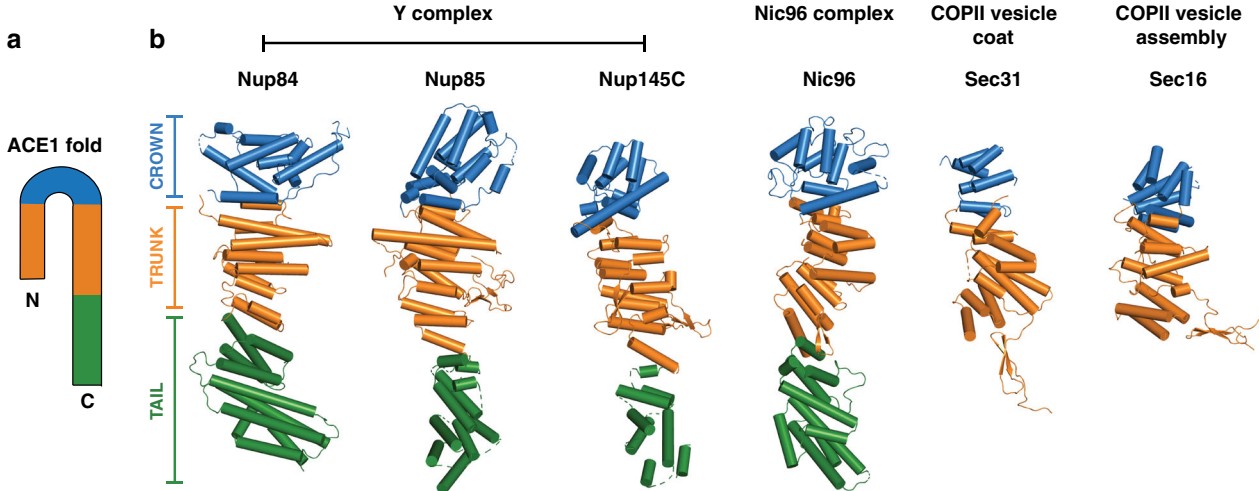

**Fig. 2 ACE1 domain proteins of the Nuclear Pore and COPII vesicle coat. a** ACE1 fold topology schematic. **b** ACE1 fold proteins. Domains indicated by color: Crown (blue), trunk, (orange), tail (green). PDB codes are as follows: Nup84 (this study), Nup85 (4XMM), Nup145C (4XMM), Nic96 (2QX5), Sec31 (2PM6), Sec16 (3MZK).

packing interfaces. In addition, Nup84 and Nup133 are conformationally flexible in solution, such that crystal packing and nanobody chaperoning together most likely only partially increase order.

Nup84 forms a continuous α-helical stack domain which, expectedly, belongs to the ACE1 architecture. Its 28 α-helices form a rectangular block with dimensions of ~124 × 37 × 46 Å, with α1–7 zig-zagging in one direction and α8–28 folding over and zig-zagging the opposite direction to the C-terminal end of the molecule (Fig. 1). The C-terminal helices interact with Nup133$_{CTD}$, which itself forms an elongated, highly arched stack of 26 α-helices (Fig. 1). The Nup84-Nup133 binding interface is compact, with ~770 Å² buried surface area, comprising α26-28 of Nup84 and α15–17 of Nup133. Typically, scaffold Nups are poorly conserved on the sequence level, despite a high degree of structural conservation[30]. However, the interface between the human homologs of Nup107-Nup133 shows a high degree of sequence conservation[21]. To see if this held true in our structure, we calculated conservation via ConSurf[41] and mapped the degree of conservation onto the surface of the molecule (Supplementary Fig. 4). To generate the surface, we added side chains in ChimeraX using the Dunbrack rotamer library[42]. From this analysis, we observe an enrichment of highly conserved residues at the interface between Nup84 and Nup133, similar to that observed in the human homologs[21] (Supplementary Fig. 4).

**Full-length Nup84 has an ACE1 fold**. With the full-length structure of Nup84, we can unambiguously detect the complete topology of the ACE1 fold. We assign the crown to α-helices 4–10, the trunk to α-helices 1–3 paired with α-helices 11–18, and the tail to α-helices 19–28 (Fig. 1). To find potentially undetected ACE1 folds in *S. cerevisiae*, we analyzed the structural homology of Nup84 by Backphyre[43] across the proteome and had relatively few hits (Supplementary Table 1). As expected, these hits included Nup145C and Nup85 from the Y complex, Nic96 from the inner ring complex, and Sec31 and Sec16 from COPII vesicle coats and assembly. However, Backphyre also identified Sea4, a component of the SEA complex, which also includes Sec13 and Seh1[44]. The SEA complex is thought to associate with the vacuolar membranes and play a role in autophagy intracellular trafficking[45]. While no high-resolution structural information is available for Sea4, it is predicted to have an N-terminal β-propeller domain, followed by an α-helical stack[44]. The Backphyre

analysis strongly suggests that Sea4 is another member of the ACE1 family.

When superimposing each ACE1 structure, one can easily detect their structural similarity (Fig. 2). Most of the trunk helices run perpendicular to the long axis of the protein, while the crown helices tip upwards. The typically longer tail helices lean downwards, away from the trunk. Comparing the tail modules across Nups, they are visibly rotated to different degrees relative to each trunk, highlighting the flexibility at the trunk-tail interface. While Sec31 and Sec16 both do not have a tail, they follow the same helical topology in the trunk and crown as the Nups.

Interestingly, VHH-SAN8 binds at the interface between the crown and trunk elements, thereby potentially rigidifying this portion of the protein (Fig. 1b). We hypothesize that this interaction may stabilize a significant crystal packing interaction between Nup84 and a symmetry related copy of Nup133 (Supplementary Fig. 5). In support of this hypothesis, previous attempts at crystallizing this complex without the nanobody yielded poorer diffracting crystals (8.3 Å), albeit in the same space group and with very similar unit cell dimensions (Table 1). With the solved structure, we could now phase these nanobody-free Nup84-Nup133$_{CTD}$ crystals. We observe poor density for the Nup84 crown, supporting our hypothesis of the nanobody stabilizing a conformationally dynamic area within Nup84 (Supplementary Fig. 6).

For future in vivo studies, we tested by size-exclusion chromatography (SEC) whether the nanobodies VHH-SAN8/9 interfere with Nup145C binding, Nup84's anchor to the Y complex. Analysis of Nup145C-Sec13-Nup84 incubated with VHH-SAN8 and VHH-SAN9 established that both nanobodies co-eluted with Nup145C-Sec13-Nup84 upon SEC (Supplementary Fig. 7). This suggests that the two nanobodies do not interfere with Nup145C binding.

**scNup133CTD is conformationally distinct from its human homolog**. Nup133$_{CTD}$ consists of 3 modules: the blade (α1– α9), the arch (α10–17), and a heel (α18–26) (Fig. 1). The first module is formed with a block of elongated helices that form a wide blade. The arch helices α10–17 are much shorter and zig-zag back and forth, curving away from the blade. The C-terminal heel domain is irregular and compact, with core helices α20-24 surrounded by α18, α19, α25, and α26. The human homolog retains much of the

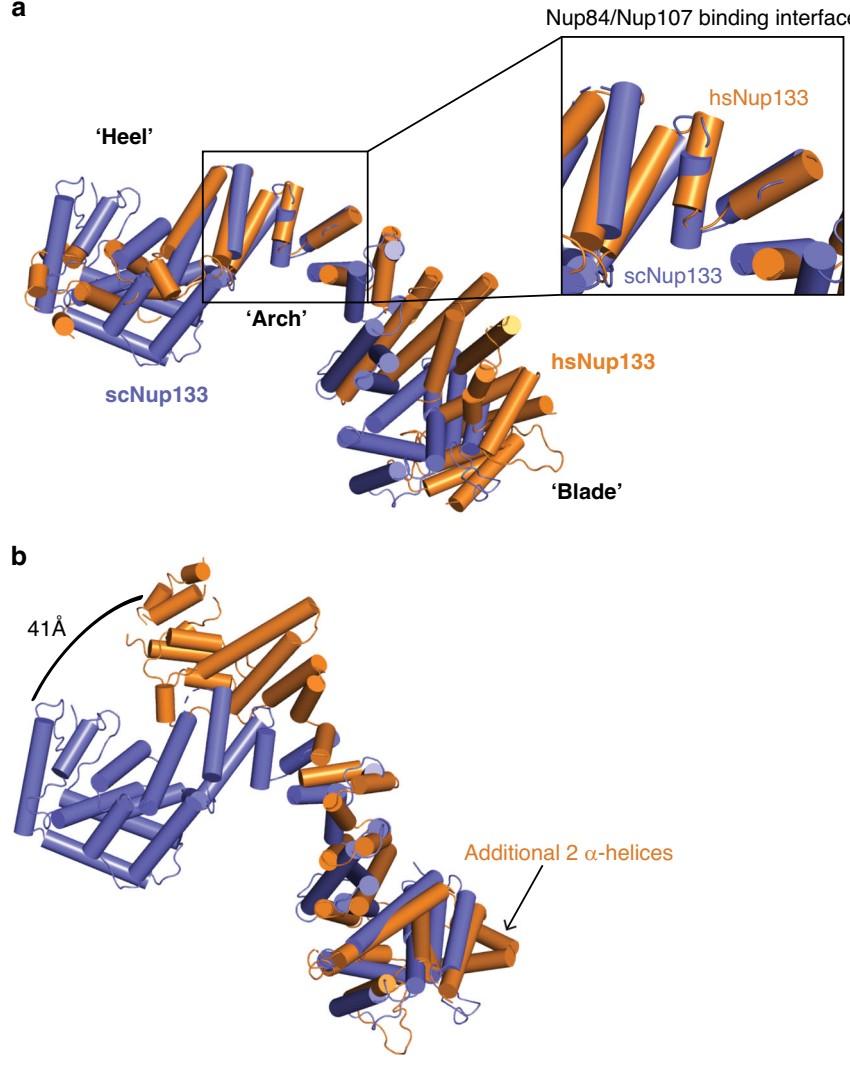

**Fig. 3 Comparison between scNup133_CTD and hsNup133_CTD.** **a** Superposition of hsNup133 (orange, PDB: 3I4R) and scNup133 (purple), using the arch element for matching. Inset zooms in on the arch where scNup84/hsNup107 bind. **b** Same superposition as in **a**, but using the blade element for matching. Arrow indicates additional helices in the hsNup133 blade domain. This superposition emphasizes the large, 41 Å conformational shift between the two heel domains.

same helical fold[21,23]. The largest difference between human and yeast Nup133 is in the C-terminal heel domain (Fig. 3a). The hsNup133 heel lacks equivalent helices to α19, α24, and α25, making it smaller than the scNup133 heel, and having a different helical topology[24]. When we superpose scNup133 and hsNup133 on the α-helices that interact with Nup84 (α15–17), we can observe a large difference between the two heel domains (Fig. 3a). However, the helical topology at the scNup84/hsNup107 binding interface overlays reasonably well, consistent with the higher degree of sequence conservation in this part of the protein (RMSD 3.5 Å over 47 Cα positions) (Supplementary Fig. 4).

The blade module of Nup133 is quite similar between yeast and human, only differing in two additional α-helices at the N terminus of the human homolog. However, the heel position relative to the arch module varies across the two structures, with the human blade module swung out ~90° relative to the yeast blade. By superposing the two blade modules, this motion becomes more apparent (Fig. 3b). This creates a swing of ~41 Å at the C terminus of the protein and rotates both the arch and heel domains the same 90° rotation. While these large conformational changes between human and yeast Nup133_CTD are likely influenced by differences in crystal packing, they are consistent

with the high degree of motion and conformational heterogeneity previously observed by negative stain EM of scNup133[25]. This rotational and translational motion, in addition to the flexible linker between Nup133_CTD and Nup133_NTD make it the most flexible Nup in the Y complex.

**Structure of the Nup133 N-terminal β-propeller bound by VHH-SAN4.** We solved the structure of the *S. cerevisiae* Nup133 N-terminal domain (Nup133_NTD, residues 55–481) at 2.1 Å resolution by molecular replacement, using the *Vanderwaltozyma polyspora* Nup133_NTD as a template[25] (Table 1, Fig. 4). Nup133_NTD is a β-propeller consisting of seven β-sheets (blades) that are radially arranged around a solvent-accessible core (Fig. 4). Blades β1–6 each contain 4 β-strands that start near the core and trace outwards. Namely, strand A is the innermost strand, where strand D is the outermost. Blade β7 is the exception, forming a 5-stranded blade made up of both, the N and C terminus, of the protein. In blade β7, the outermost strands D and E are from the N-terminal part of the β-propeller and strands A-C are from the C-terminal portion. This Velcro closure is common in many β-propeller domains and is thought to stabilize the closed fold[46]. In comparison to the Nup133_NTD structures from

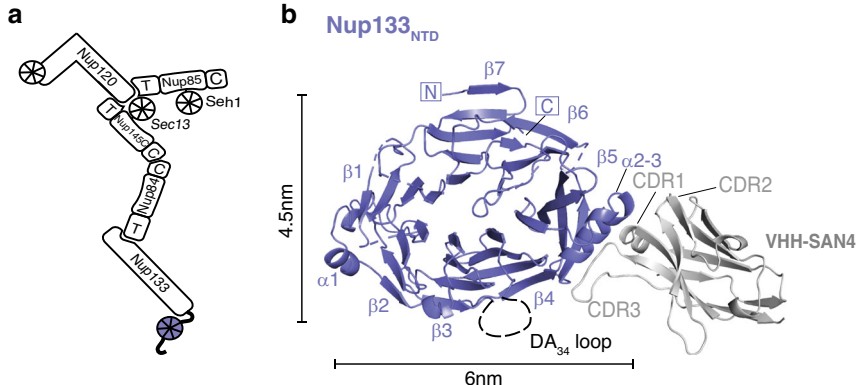

**Fig. 4 Structure of Nup133$_{NTD}$-VHH-SAN4 complex. a** Schematic of the Y complex. The region included in the structure is colored (purple—Nup133). Elements of ACE1 fold proteins are indicated: T—tail and C—crown adjacent to the central trunk element. **b** Nup133$_{NTD}$-VHH-SAN4, with Nup133$_{NTD}$ in purple and VHH-SAN4 in light gray. N and C termini are indicated, and β-sheets and α-helices are labeled on Nup133$_{NTD}$. Complementarity determining region (CDR) loops are labeled on VHH-SAN4.

other species, the domain is structurally highly similar to *V. polyspora* (RMSD 1.36 Å over 381 Cα positions) and quite similar to human (RMSD 3.06 Å over 377 Cα positions) (Fig. 5a). All three β-propellers have seven blades with two significant elaborations, an α-helical hairpin between blades 4 and 5 (α2–3) and a 20-residue, mostly disordered loop between strand 3D and strand 4 A (DA$_{34}$ loop) (Fig. 4). Additionally, this structure has an elongated loop between stands A and B in blade β1 that is ordered due to crystal packing contacts. The core structure of Nup133$_{NTD}$ is conserved across the three species.

The nanobody, VHH-SAN4, binds along strand 5D, via its large CDR3 loop (Fig. 4). The opposite end of the nanobody facilitated packing with three symmetry related copies of Nup133$_{NTD}$, critical for lateral assembly within the crystal. Additionally, we solved the structure of Nup133$_{NTD}$ bound by a second nanobody, VHH-SAN5, to 2.8 Å resolution (Table 1 and Supplementary Fig. 8). VHH-SAN5 recognizes an adjacent epitope to VHH-SAN4, also binding along strand 5D using primarily its CDR3. There are additional contacts between CDR2 and the α-helical hairpin α2–3. The N-terminal β-strand of VHH-SAN5 also makes extended contact with strand D of blade β6 on an adjacent copy of Nup133$_{NTD}$, ultimately forming a hexameric packing assembly in the core of the asymmetric unit (Supplementary Fig. 8). Two such Nup133$_{NTD}$-VHH-SAN5 heterohexamers stack, creating the striking dodecameric asymmetric unit (Supplementary Fig. 8). Despite the difference in packing between the two structures, the six copies of Nup133$_{NTD}$ in the VHH-SAN5 bound structure are highly similar both to each other (RMSD 0.17–0.22 Å) and to the VHH-SAN4 bound structure (RMSD 0.68–0.81 Å). The majority of the differences come in conformations of multiple, extended loops that are generally more ordered in VHH-SAN4 bound Nup133$_{NTD}$, presumably because of better crystal packing. Specifically, the loops containing residues 72–86, 143–160, and 404–412 are disordered in the majority of VHH-SAN5 bound Nup133$_{NTD}$ copies along with the α-helical hairpin α2–3 (residues 176–190) that are visible in the VHH-SAN4 bound map. On the other hand, loop residues 430–440 are better ordered in the VHH-SAN5 bound structure.

**Yeast Nup133 has an ALPS motif.** After observing the same disordered DA$_{34}$ loop in our structure that is present in the human Nup133$_{NTD}$, we wondered if the yeast Nup133 also has a functional ArfGAP1 lipid packing sensing (ALPS) motif. A computationally determined ALPS motif has been identified in scNup133$_{NTD}$ via homology modeling[25]. However, the hydrophobic moment of

this loop is weaker than the human DA$_{34}$ loop due in part to the presence of an asparagine and lysine within the hydrophobic half of the amphipathic helix (Fig. 5a). The increase in hydrophilicity in the yeast Nup133$_{NTD}$ DA$_{34}$ loop led to earlier speculations that this feature may not be conserved between metazoa and yeast[22,37]. To directly test whether yeast Nup133$_{NTD}$ has a functional ALPS motif, we performed liposome floatation assays. This assay determines whether a protein interacts with liposomes through the floatation of a liposome-protein mixture to the top of an iso-osmotic gradient during ultracentrifugation[47]. We observed that WT-Nup133$_{NTD}$ floats with liposomes comprising yeast polar lipids, but pellets without liposomes (Fig. 5b). In contrast, the Nup133$_{NTD}$ΔALPS mutant, in which we replaced the DA$_{34}$ loop with an isosteric linker (GGGGSGGGGS), did not interact with liposomes.

We also wondered if we could visualize any remodeling of liposomes by negative stain electron microscopy, as other yeast Nups (Nup1, Nup60, Nup53) with ALPS motifs have done to various extents[48,49]. Indeed, WT-Nup133$_{NTD}$ produces small fringe-like protrusions on the surface of liposomes (Fig. 5c). This is in contrast to human Nup133$_{NTD}$, which did not alter liposomes[48]. However, this difference could be due to the varying liposome composition and size across these experiments. With the ALPS deletion mutant, Nup133$_{NTD}$ΔALPS, these protrusions did not occur (Fig. 5c). The results of these disparate assays demonstrate that the yeast Nup133$_{NTD}$ has an ALPS motif in its DA$_{34}$ loop, a conserved motif with human Nup133, that can bind to and modify curved biological membranes.

**A complete composite model of the yeast Y complex.** With our additional structures in hand, we constructed a composite model of the yeast Y complex. The assembly of this model was straightforward, as both, the structure we describe here and the previously published hub structure, overlap significantly within Nup84, and they superpose well (RMSD 2.1 Å over 454 Cα positions). While the structures of both, the Nup84-Nup133$_{CTD}$ element and the Y complex hub, are of modest resolution, many fragments of the assembly are known at much higher resolution. Our previous model of the Y complex comprised structures solved from *S. cerevisiae*, but it also contained homology-based models based on structures from other species[26]. In the case of Nup84, where no full-length structure was known, we used Nic96, another ACE1 domain protein with a high-resolution structure, to model the missing four-helix element. In comparing our new composite model to our previous model[26], we observe a number

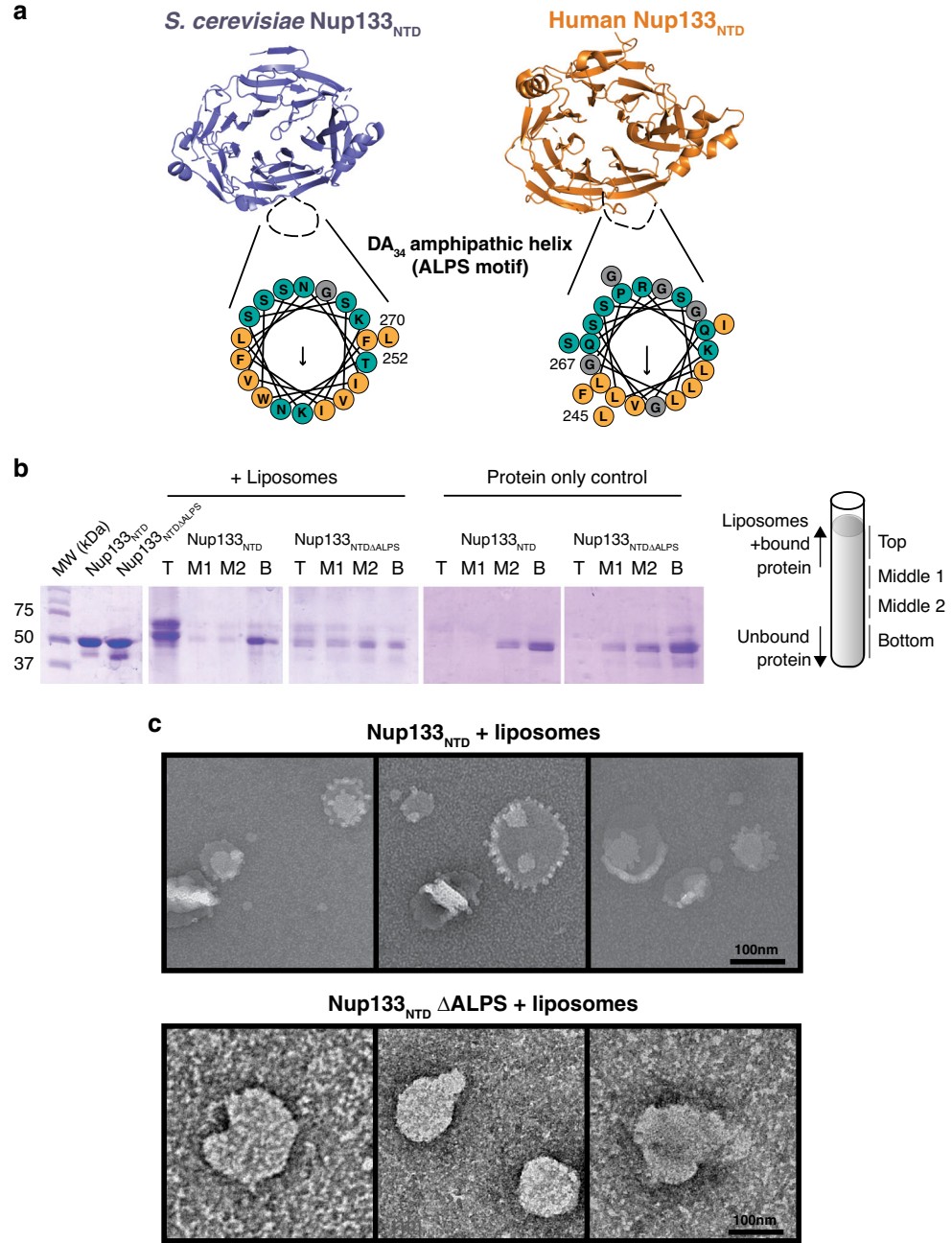

**Fig. 5 Nup133$_{NTD}$ has an ALPS motif. a** Structures of scNup133$_{NTD}$ (purple) and hsNup133$_{NTD}$ (orange). The missing DA$_{34}$ loop in the structures are indicated by a black dotted line. Helical wheel diagram for each loop is shown with polar residues (teal), nonpolar (orange), and glycine (gray) colored. Arrow indicates predicted hydrophobic face and length of arrow scales with strength of the hydrophobic moment. **b** Liposome floatation assay with Nup133$_{NTD}$. Panels show Coomassie stained SDS-PAGE fractions isolated from the gradients. A cartoon of fractions is shown on the right. **c** Negative stain electron microscopy images of liposomes preincubated with Nup133$_{NTD}$ (upper) and Nup133$_{NTD}$ ΔALPS (lower). Scale bar at the bottom left of each panel. Panels are representative images taken from two independent experiments.

of conformational differences (Fig. 6). Both models have the same dimensions overall, but superpositioning the two models using Nup145C, the most rigid part of the Y complex, reveals hinge points in both, the short arms and the stalk of the Y. In the short arm of Nup85-Seh1, we notice significant motion around the trunk-to-tail transition of Nup85. Additionally, Nup120 is bent more outward from the hub in our previous model. The stalk has multiple points of flexibility. The Nup84 tail is positioned about ~4 nm away from its previous position, with the majority of the movement occurring around the hinge at the trunk-tail interface. This motion causes the Nup133 heel domain to shift ~3 nm. The

Nup133 arch between the heel and the blade moves the blade domain an additional ~3 nm, with the motion from the long stalk totaling ~7 nm. Measuring motion between the Nup133$_{NTD}$ and Nup133$_{CTD}$ seems pointless, as the β-propeller is tethered by a flexible, unstructured element to the α-helical stack[25]. While both models provide snapshots of Y complexes dictated by the packing environment of the underlying crystal structures, electron microscopy studies of the Y complex support the notion of defined motions within the complex[28,29]. Overall, these two composite models capture different conformations, highlighting the range of motion and the hinge points of the Y complex.

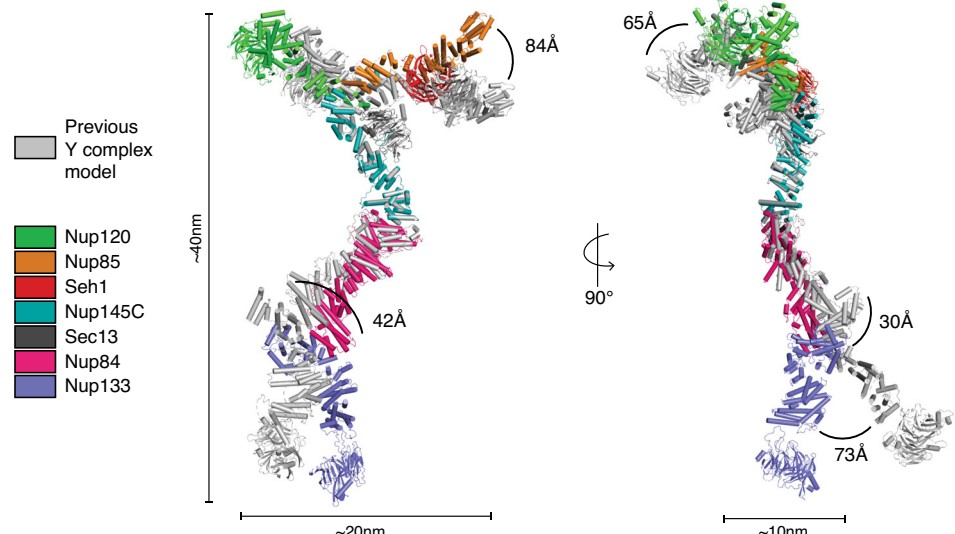

**Fig. 6 Composite model of the Y complex.** Composite model of the *S. cerevisiae* Y complex super-positioned with the previous *S. cerevisiae* composite model[26], using Nup145C for matching. Colors of each nup are indicated in the key on the left. Overall dimensions of the Y complex are indicated. Conformational differences between the models are shown by arcs between the same nups, with the distance of each listed.

**The composite Y complex model in the NPC assembly**. While our new composite model represents one snapshot of the Y complex conformation, we tested how well it could be positioned into the recently published cryo-electron tomography (cryo-ET) reconstruction from *S. cerevisiae*[4]. Our model of the Y complex fit quite well into the density for both the cytoplasmic and nucleoplasmic rings of the NPC (Fig. 7). The structure of Nup84-Nup133$_{CTD}$ reported here is flatter and straighter than previous models suggested[26,33] (Fig. 6), resulting in a better fit for our revised composite model. The Nup133$_{NTD}$ is only approximately positioned relative to the Nup133$_{CTD}$, as the connection between the two is highly flexible. However, Nup133$_{NTD}$ is positioned with its ALPS motif adjacent to the NE. The fit for the nucleoplasmic ring was slightly better, suggesting some conformational differences between the Y complexes on the cytoplasmic and nucleoplasmic faces of the NPC (Fig. 7b, see Materials and Methods). The difference between the cytoplasmic and nucleoplasmic Y complex conformation is more striking in docking attempts using our previous composite model (Fig. 7). While the fitting correlation score and positioning for the cytoplasmic ring was comparable between both models, the previous model fit much more poorly to the nucleoplasmic ring (see Materials and Methods). The overall shape and dimensions of this new composite model are more consistent with this cryo-ET reconstruction.

## Discussion

Here we describe two structures, Nup84-Nup133$_{CTD}$ and Nup133$_{NTD}$, that complete the structural inventory for the *S. cerevisiae* Y complex. This revised composite model pushes forward our molecular understanding of the greater NPC structure, which will ultimately give us a better molecular understanding of the many functions of the NPC. Complementary to our bottom-up structural approach studying Nup subassemblies, studies on the massive assembly of the NPC from a multitude of species by cryo-ET yielded density maps at a range of resolution. In yeast, there are two recent cryo-ET reconstructions. One analyzed NPCs after detergent extraction from the NE, with a local resolution of about ~7 nm in the cytoplasmic and nucleoplasmic rings, precluding any reasonable attempt at docking the Y complex[33]. A second study describes a cryo-ET reconstruction from intact yeast nuclei to ~2.5 nm resolution[4].

Both of these in situ docking studies have docked single Nups or small Nup assemblies, as the previous Y complex models did not fit well as a rigid body. However, our revised composite model fit quite well into the higher resolution cryo-ET map (Fig. 7)[4]. There still remains room for improvement in both the resolution of the cryo-ET map to secondary structure resolution and the accuracy of the model for the Y complex. Such improved resolution could, for example, reveal the molecular details for the differences between Y complexes forming two reticulate double-rings in vertebrates, while only a single ring in *S. cerevisiae*. It will also be fundamentally important to understand whether all NPCs within one cell, or across different cell types, are identical, or whether there are populations with different conformation and/or composition. The well-documented modularity of the NPC suggests the latter[5,9,34,50]. For *S. cerevisiae*, having a composite model of the Y complex that does not rely on homology modelling, should help in answering these more detailed questions on the NPC architecture going forward. Homology modelling leaves a level of uncertainty, especially when the proteins in question have very low sequence identity, which is intrinsic to most scaffold nucleoporins.

Now that we have a complete Y complex and multiple structures for most of the Nups, we can delineate the flexible elements within the structure. We can also now see the conformational differences in the Y complex on different faces of the NPC (Fig. 7). These differences were also noted in a cryo-ET study on *Xenopus laevis* NPCs, suggesting this difference as a conserved feature of the assembly[3]. Future studies can employ techniques such as molecular dynamics, moving the molecule at known hinge points to fit into the map, rather than individual nups that look highly similar at >10 Å resolution, to ultimately get a high-resolution assembly model. Overall, the structures presented here provide additional elements for better constructing composite NPC structures combining X-ray crystallographic and electron microscopy methods.

The observation of the great flexibility of the Y complex has led many to speculate its importance in NPC biology. One key contributor to this flexibility is the ACE1 fold in multiple Y complex nups. The structurally related Sec16 and Sec31 of the COPII vesicle coat both need to coat dynamically shaped, highly curved membranes through a vesicle's life cycle[51]. While the Y

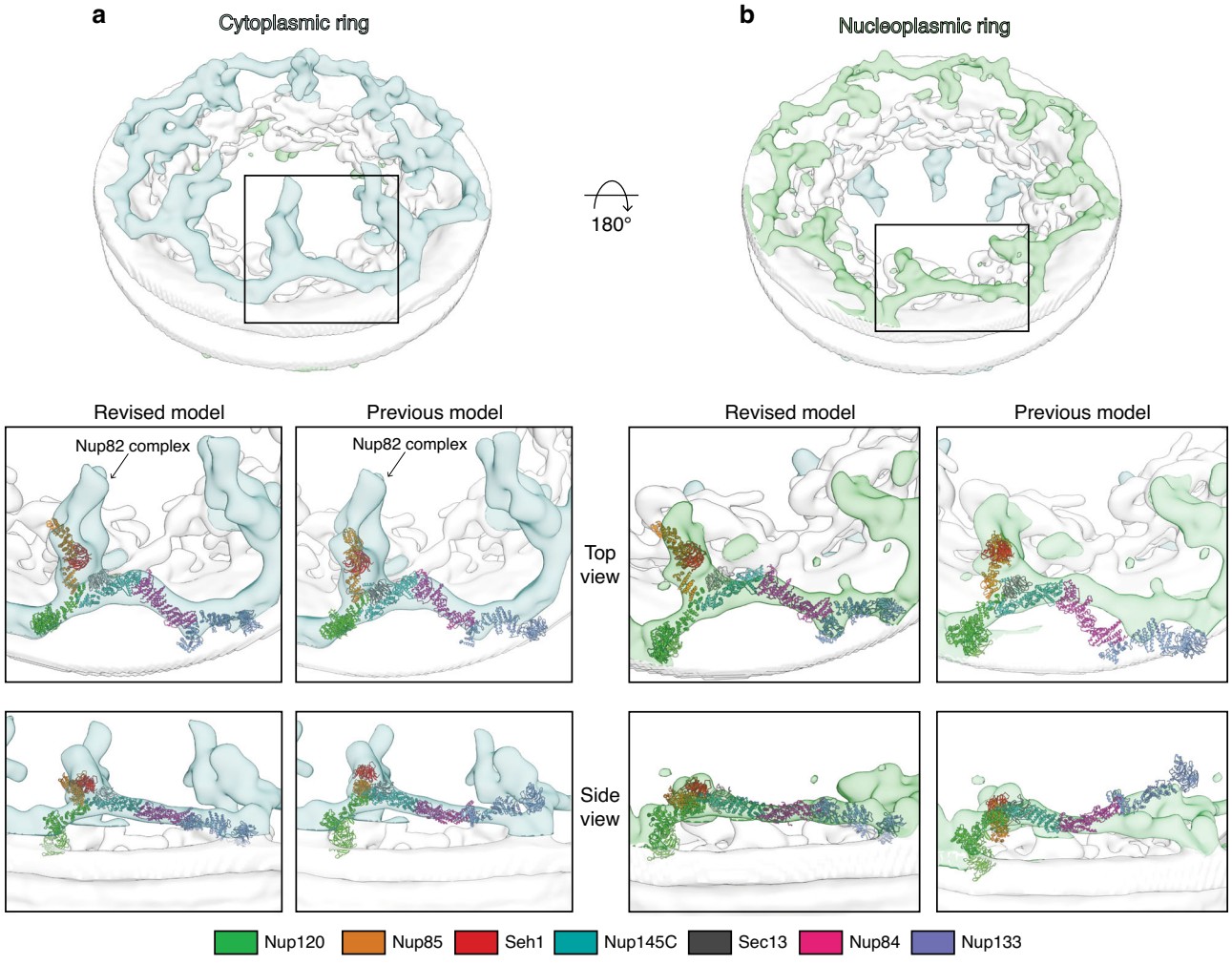

**Fig. 7 Fitting of the composite *S. cerevisiae* Y complex into the cryo-ET map of the NPC.** Cryo-ET map from *S. cerevisiae* (EMD-10198) with the density corresponding to the **a** cytoplasmic ring of the NPC colored in light blue and **b** the nucleoplasmic ring colored in light green. The top-scoring global fit for both the previous (left panel) and revised (right panel) composite *S. cerevisiae* Y complex models shown in both a top and side view. Y complex nups colored as in legend below.

complex arrangement within the NPC differs from that of the COPII vesicle coat, the conservation of the ACE1 fold and its inherent flexibility could be critical for NPC assembly and integrity. NPC heterogeneity in both size and constituents has been suggested by a number of studies, and therefore it is likely the core scaffold of the NPC would need to accommodate these variances[34,52–55]. The NPC and Y complex may also have to be flexible in order to adapt to a variety of cargoes. The NPC can facilitate the transport of surprisingly large cargoes, up to ~39 nm in diameter, such as intact Hepatitis B viral capsids[56,57]. Conceptually, there is no need for the nucleoplasmic and cytoplasmic rings to adopt to cargo traveling the central channel, since the ring opening is much larger than even the biggest cargoes. However, integral inner nuclear membrane proteins pass through the NPC presumably via peripheral channels adjacent to the NE[58], unless they are inserted post-translationally[59]. Some of these proteins, such as Heh1 and Heh2 in *S. cerevisiae*, have significantly large nucleoplasmic domains that would need to pass near to the scaffold of the NPC as they translocate[60,61]. As the main scaffolding component of both the cytoplasmic and nucleoplasmic rings, it is still an unsolved problem if and how the cyto- and nucleoplasmic rings are involved in this process. It is easy to imagine that the observed Y complex flexibility plays a part in this translocation process.

Another potential for required flexibility could be in the assembly of new NPCs. The Y complex is one of the first complexes present during NPC assembly[62,63]. Knowing that Nup133 can anchor into the nuclear envelope with its ALPS motif, it is not unlikely that the Y complex ring of a newly assembling NPC would need to change shape as the inner and outer nuclear membranes transition through the extreme curvatures required for NE fusion during the building of a new NPC[38]. This would be especially critical in assembly into a pre-formed NE during interphase or a closed-mitotic system, such as in *S. cerevisiae*.

The conformation of the yeast Nup133 ALPS motif also raises many questions as to its relevance in vivo. Various deletions of Nup133, including both a complete deletion and deletions to portions of its N terminus result in decreased fitness in yeast and a clustering phenotype of the NPC on the nuclear envelope[64–66]. The mechanism and effects on the NPC structure and assembly during clustering is still unknown. Deletion of only the $DA_{34}$ loop has only been done in humans and was found to be essential for interphase assembly[38]. Careful dissection of the role of the Nup133 ALPS motif could answer questions on the conservation of NPC assembly across yeast and human. Confirming the presence of the ALPS motif on Nup133 puts the $Nup133_{NTD}$ in immediate proximity to the membrane. Again, this β-propeller domain has been separately docked in situ, but its orientation was

unclear[31]. With knowing the relative position of Nup133, one can begin to examine the head-to-tail interaction of neighboring Y complexes in greater detail.

## Methods

**Construct generation.** Nup133$_{NTD}$ (aa55-481), Nup133$_{CTD}$ (aa521-1157), and Nup84 were cloned from *S. cerevisiae*. Nup133$_{NTD}$ was N-terminally fused with a human rhinovirus 3 C (3 C) protease cleavable 6xHis tag and cloned into an ampicillin resistant, T7-promoter-based bacterial expression vector. We cloned the coding sequences for the heterodimeric complex of Nup84 and Nup133$_{CTD}$ into a T7-promoter-based bicistronic bacterial expression vector with ampicillin resistance. Nup84 was N-terminally fused with a 3C cleavable 10xHis-8xArg-SUMO tag. Nup133$_{CTD}$ was C-terminally fused with an 8×Arg tag. The Nup133$_{NTD}$ ALPS mutant was generated by replacing residues 252–270 with GGGGSGGGS by inverse PCR. Upon VHH selection by phage display and ELISA, VHH-SAN4, 5, 8, and 9, were sub-cloned for expression. Each VHH sequence was N-terminally fused with a 14×His bdSUMO tag[67] and cloned into a T7-promoter-based bacterial expression vector with ampicillin resistance. The Nup145C-Sec13fusion-Nup84$_{1–424}$ construct was previously described in ref. [18]. Primers used in cloning are listed in Supplementary Table 2.

**Protein expression and purification.** The Nup84-Nup133$_{CTD}$ vector was transformed into *Escherichia coli* LOBSTR-RIL(DE3) (Kerafast)[68] cells and protein production was induced with 0.2 mM IPTG at 18 °C for 12–14 h. Nup133$_{NTD}$ was expressed separately under identical conditions. Cells were collected by centrifugation at 6000 × g, resuspended in lysis buffer (50 mM potassium phosphate (pH 8.0), 500 mM NaCl, 30 mM imidazole, 3 mM β-mercaptoethanol (βME), 1 mM PMSF) and lysed using a high-pressure cell homogenizer (Microfluidics LM20). The lysate was cleared by centrifugation at 12,500 × g for 25 min. The soluble fraction was incubated with Ni Sepharose 6 Fast Flow beads (GE Healthcare) for 30 min on ice. After washing the beads with lysis buffer, the protein was eluted (250 mM imidazole, pH 8.0, 150 mM NaCl, 3 mM βME).

Nup84-Nup133$_{CTD}$ was dialyzed into 10 mM HEPES/NaOH pH 8.0, 100 mM NaCl, 0.1 mM EDTA, and 1 mM DTT and loaded directly onto a 5 ml HiTrap SP Sepharose fast flow column (GE Healthcare). Bound Nup84-Nup133 was eluted from the column using a 15 column volume (CV) gradient of 10 mM HEPES/ NaOH pH 8.0, 50 mM–1 M NaCl, 0.1 mM EDTA, and 1 mM DTT. The pooled eluate was loaded onto a Superdex 200 26/60 gel filtration column equilibrated in 10 mM Tris/HCl pH 7.5, 150 mM NaCl, 0.1 mM EDTA, 1 mM DTT. The Nup84-Nup133 peak was pooled and concentrated.

Nup133$_{NTD}$ was dialyzed into 10 mM Tris/HCl pH 7.5, 150 mM NaCl, 0.1 mM EDTA, 1 mM DTT. Nup133$_{NTD}$ was then concentrated and loaded on a Superdex 75 16/60 gel filtration column equilibrated in the same buffer. The Nup133$_{NTD}$ peak was pooled and concentrated. The Nup133$_{NTD}$ ALPS mutant was prepared identically.

All VHH constructs were transformed, grown, harvested, and lysed as above. After lysis, the soluble fraction was incubated with Ni Sepharose 6 Fast Flow beads (GE Healthcare) for 30 min on ice. The beads were then washed with lysis buffer and transferred to low imidazole buffer (50 mM potassium phosphate pH 8.0, 500 mM NaCl, 10 mM imidazole, 3 mM βME) along with 10 μg bdSENP1[67] (bdSUMO protease) and incubated 2 h at 4 °C. Then the flow through containing the cleaved VHHs was collected, along with a 2 CV wash with low imidazole buffer. Cut tags and uncut protein was eluted as above. Each VHH was concentrated and loaded onto a Superdex 75 16/60 gel filtration column equilibrated in 10 mM Tris/HCl pH 7.5, 150 mM NaCl, 0.1 mM EDTA, 1 mM DTT. Each VHH peak was pooled and concentrated.

**VHH library generation and selection by phage display.** Alpaca immunization, library generation, and M13 phage generation was done as previously described[69]. The animal was immunized against recombinantly expressed full-length Y complex (Nup120-Nup85-Seh1-Nup145C-Sec13-Nup84-Nup133). VHHs were selected by panning against Nup84 and Nup133$_{NTD}$. The panning selection and ELISA protocols were done as previously described (see accompanying paper, https://doi.org/10.1101/2020.06.19.161075).

**Protein crystallization.** Initial crystals of Nup84-Nup133$_{CTD}$-VHH-SAN8 were obtained at 18 °C in 2 days as part of the Morpheus HT suite (Molecular Dimensions) in a 96-well sitting drop tray with a reservoir containing 12% (v/v) ethylene glycol, 6% (w/v) PEG 8,000, 0.1 M imidazole titrated with MES pH 6.5, 30 mM sodium nitrate, 30 mM sodium phosphate dibasic, 30 mM ammonium sulfate. Sitting drops of 2 μl protein at 8 mg/ml and 1 μl of precipitant (5–6% PEG 8,000, 30–32.5% ethylene glycol, 0.1 M imidazole titrated with MES pH 6.5, 15 mM sodium nitrate, 15 mM sodium phosphate dibasic, 15 mM ammonium sulfate) produced crystals in 3 days, which continued to grow over the following week. Nup84-Nup133$_{CTD}$ + VHH-SAN8/9 crystallized under identical conditions. Crystals were harvested without additional cryo-protectant. Derivative crystals were obtained by applying 0.2 μl of 0.1 M [TeW$_6$O$_{24}$]$^{6-}$ (MiTeGen) to drops

containing Nup84-Nup133$_{CTD}$-VHH-SAN8 crystals. Crystals were harvested after 1 hour and cryo-cooled in liquid nitrogen.

Initial crystals of Nup133$_{NTD}$-VHH-SAN4 were obtained at 18 °C in 14 days as part of the IndexHT suite (Hampton Research) in a 96-well sitting drop tray with a reservoir containing 25% PEG 3,350, 0.1 M Bis-Tris/HCl pH 5.5, 0.2 M ammonium sulfate. Hanging drops of 1 μl protein at 7 mg/ml and 1 μl precipitant (23–24% PEG 3,350, 0.2 M Ammonium sulfate, 0.1 M Bis-Tris/HCl pH 5.5) produced diffraction-quality crystals of rod-shaped clusters in 12 days to 1 month. Crystals were transferred into a cryo-protectant solution containing the crystallization condition with 12% PEG 200 and cryo-cooled in liquid nitrogen.

Initial crystals of Nup133$_{NTD}$-VHH-SAN5 were obtained at 18 °C in 1 day as part of the Protein Complex suite (Qiagen) in a 96-well sitting drop tray with a reservoir containing 15% PEG MME 2,000, 0.1 M potassium chloride, 0.1 M Tris/ HCl pH 8.0. Hanging drops of 1 μl protein at 5 mg/ml and 1 μl precipitant (11% PEG MME 2000, 0.2 M potassium chloride, 0.1 M Tris/HCl pH 8.0) produced diffraction-quality diamond-shaped crystals in 1–3 days. Crystals were transferred into a cryo-protectant solution containing the crystallization condition with 12% PEG 200 and cryo-cooled in liquid nitrogen.

**Data collection and structure determination.** Data collection was performed at the Advanced Photon Source end station 24-IDC. All data processing steps were carried out with programs provided through SBgrid[70]. Data reduction was performed using HKL2000[71] and XDS[72]. Statistical parameters of data collection and refinement are given in Table 1. Structure figures were created in PyMOL (Schrödinger LLC).

For Nup84-Nup133-VHH-SAN8, the structure was solved with single anomalous scattering using a [TeW$_6$O$_{24}$]$^{6-}$ (MiTeGen) derivative dataset. One cluster site was found with SHELXC/D/E from the CCP4 suite[73]. *AutoSol* in the PHENIX suite was used to refine the site and generate an initial map[74]. The map was iteratively refined using *phenix.refine* after placement of poly-alanine helices in *Coot*[75]. Once the density had improved, known fragment of Nup133 heel (PDB: 3KFO) was placed via *Dock in Map* in Phenix. After one round of refinement, a known fragment of Nup84 (PDB: 3JRO) was placed in the map via molecular replacement. The remaining parts of the proteins were manually built, using the homologous human structure as a guide (PDB: 3I4R). The asymmetric unit contains one Nup84-Nup133-VHH-SAN8 complex. *Phenix.rosetta_refine*[76] was employed once all of the density for Nup84 and Nup133 was occupied. Only rigid body refinement was done on the previously solved parts of the molecule, along with applying secondary structure restraints throughout the refinement process. Near the end of refinement, TLS parameters were used. Since the complementarity determining region (CDR) loops were unclear in the map, a generic nanobody with its CDR loops removed was used as a rigid body (PDB: 1BZQ)[77]. The model was built without side chains, as no clear density for them can be observed in the electron density map. The sequence has been applied to the model based on the docked, previously solved fragments and alignment with the homologous human structure of Nup133 (PDB: 3I4R).

The structure of Nup133$_{NTD}$-VHH-SAN5 was solved by molecular replacement (MR) using *Phaser-MR* in PHENIX[74]. A two-part MR solution was obtained by sequentially searching with models of Nup133$_{NTD}$ and VHH-SAN5. For Nup133$_{NTD}$, we used the structure of *Vanderwaltozyma polyspora* Nup133$_{NTD}$ (PDB: 4Q9T, 45% sequence identity). For VHH-SAN5, we used a nanobody structure with its CDR loops removed (PDB: 1BZQ). The asymmetric unit contains six copies of Nup133$_{NTD}$ + VHH-SAN5. The structure of Nup133$_{NTD}$-VHH-SAN4 was solved by MR using *Phaser-MR* in PHENIX, just as Nup133$_{NTD}$-VHH-SAN5. However, we used our previously solved Nup133$_{NTD}$ as the search model. The asymmetric unit contains one copy of Nup133$_{NTD}$ + VHH-SAN4. All manual model building steps were carried out with Coot and *phenix.refine* was used for iterative refinement. Near the end of refinement for Nup133$_{NTD}$ + VHH-SAN5, TLS parameters were used.

**Liposome preparation.** To prepare liposomes, a thin-film of yeast polar lipid extract (YPL, Avanti Polar Lipids) at 10 mg/ml in chloroform was dried in a glass vial under nitrogen gas, washed twice with anhydrous pentane (Sigma–Aldrich) and resuspended overnight at 10 mg/ml in 10 mM Tris/HCl pH 7.5, 150 mM NaCl, 0.1 mM EDTA, 1 mM DTT. Following resuspension, three freeze-thaw cycles were completed. To obtain unilamellar liposomes, extrusion (Avanti Mini-extruder) was completed with 21 passes through a 0.03 μM polycarbonate membrane (Avanti Polar Lipids). Liposomes were stored at 4 °C and used within a week.

**Liposome floatation assay.** Liposomes preparations and protein purification were completed as detailed above. A total of 1 μg protein was incubated with 500 μg yeast polar lipids in 50 μl of floatation buffer (100 mM potassium acetate, 2 mM sodium acetate, 20 mM HEPES/NaOH pH 7.4, 2 mM DTT) for 30 mins at room temperature. After incubation, the 50 μl liposome sample was mixed with 644 μl 48% Nycodenz (5-(N-2,3-dihydroxypropylacetamido)-2,4,6-tri-iodo-N,N-bis(2,3-dihydroxypropyl)isophthalamide, Accurate Chemical) solution in floatation buffer and applied to the bottom of a 11 × 34 mm ultracentrifugation thick-wall tube. A 1.524 ml layer of 40% nycodenz solution was layered above, followed by a 592 μl layer of floatation buffer. Samples underwent ultracentrifugation in an SW55Ti

rotor for 2 h at 4 °C, 240,000 × *g* in an Optima Max Ultracentrifuge (Beckman). Following ultracentrifugation, four fractions were taken, top (832 µl), middle 1 (566 µl), middle 2 (566 µl), and bottom (844 µl). Protein was extracted from the fractions by methanol chloroform precipitation. Analysis of the assay was completed by a 12% SDS-PAGE visualized with Coomassie stain.

**Negative stain electron microscopy.** Nup133$_{NTD}$-liposome mixtures at 1 mg/ml were loaded on glow-discharged (EMS 100, Electron Microscopy Sciences) continuous carbon film grids (CF200-Cu, Electron Microscopy Sciences). After 45 s of adsorption on grids, the samples were blotted with Whatman filter paper and the specimen on the grid was immediately stained with 2% w/v uranyl acetate for 30 s. The specimen was blotted, stained once more, reblotted, and air dried. Electron micrographs were recorded on a FEI Tecnai Spirit BioTwin microscope (FEI) operated at 80 keV and equipped with a tungsten filament and an AMT XR16 CCD detector.

**Fitting the composite Y complex model into the cryo-ET map of the *S. cerevisiae* NPC.** A global fitting approach was performed using our composite model of the Y complex into the cryo-ET map (EMD-10198) from *S. cerevisiae*[4]. Density corresponding to the cytoplasmic and nucleoplasmic rings alone were conservatively defined as subregions in Chimera to eliminate positioning in the nuclear envelope or inner ring, respectively. Docking was done with Fitmap in Chimera, with 10,000 placements with an apparent resolution of 30 Å for the Y complex. Scoring was done as correlation around zero. The highest scoring solution for the cytoplasmic placement had a correlation score of 0.68 (Fig. 7a). The highest scoring solution for the nucleoplasmic placement had a correlation score of 0.73 (Fig. 7b). Fitting of our previous model of the Y complex[26] had a correlation score of 0.66 in the cytoplasmic ring and 0.61 in the nucleoplasmic ring (Fig. 7).

**Reporting summary.** Further information on research design is available in the Nature Research Reporting Summary linked to this article.

## Data availability

Coordinates and structure factors have been deposited in the Protein Data Bank under accession codes PDB "6X02" (Nup84-Nup133$_{CTD}$-VHH-SAN8), "6X03" (Nup84-Nup133$_{CTD}$-VHH-SAN8/9), "6X04" (Nup133$_{NTD}$-VHH-SAN5), "6X05" (Nup133$_{NTD}$-VHH-SAN4). The cryo-ET map used for fitting the composite model of the Y complex is described elsewhere[4] and available from the Electron Microscopy Data Bank (EMDB) under accession number "EMD-10198".

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

## Acknowledgements

We thank Alexander Ulrich for generating initial expression constructs and preliminary crystals for the Nup84-133 complex. Research was supported by the NIH under grant number R01GM77537 (T.U.S.) and T32GM007287 (S.A.N.) The X-ray data collection was conducted at the APS NE-CAT beamlines, supported by award GM103403 from the NIH. Use of the APS is supported by the US Department of Energy, Office of Basic Energy Sciences, under contract no. DE-AC02-06CH11357.

## Author contributions

S.A.N. and T.U.S. designed the study. S.A.N. performed the experiments. D.L.T. assisted with the liposome interaction experiments. S.A.N. and T.U.S analyzed and interpreted the data, and wrote the manuscript with input from D.L.T.

## Competing interests

The authors declare no competing interests.
