## [Peer Review File · Nature Communications]

Review of two back-to-back (?) papers of the Schwartz lab

I have read two papers of the two back-to-back papers of the Schwartz lab with great interest. I recommend publication of the first one, entitled 'Structure of the yeast Nup84-Nup133 complex 1 details flexibility and reveals universal conservation of the membrane anchoring ALPS motif' because the data are technically sound, the paper provides strong evidence for its conclusions, the results are novel and the manuscript is important to scientists in the specific field.

Minor comments:

- Line 96. The authors solved a low resolution structure (6.4Å) of the Nup84-Nup133 C-terminal domain in complex with VHH-SAN8 but give no single argument why this is the best they could do?
- Line 112. The authors also solved a low resolution structure (7.4Å) of the Nup84-Nup133 C-terminal domain in complex with VHH-SAN8 and VHH-SAN9 but give no single argument why this is the best they could do?
- Line 162. The authors discuss the phasing of nanobody-free Nup84-Nup133CTD crystals but do not show the data? As these data are used to support an important hypothesis, it would make sense for the experts in the NPC field to describe these results in the supplementary data?
- Line 184. Does it make sense to talk about an RMSD with an accuracy beyond the comma if you start from 6-7Å resolution structures? On line 258, the RMSD is only given with one figure beyond the comma.
- Line 298: What is 'this new model' referring to? 'Two new structures' or 'structural inventory'?
- Line 746. Please add the resolution of the structure to the legend and indicate the PDB code.
- Line 766. Add the PDB code to scup133
- Line 772. Add PDB codes
- Line 782. Add PDB codes

However, I have my doubts about the accompanying paper entitled 'A nanobody suite for yeast scaffold nucleoporins 1 provides details of the Nuclear Pore Complex structure'. Although the data are technically sound, the whole manuscript reads like an 'ennobled' materials and methods section with entire result sections and figures that normally fit the materials and methods section and the supplementary figures of recent high impact papers:

- Result section 'A nanobody library to the Y complex and Nic96' describes the selection of Nanobodies, following standard procedures
- Result section 'Nanobodies bind with varying kinetics, but strong affinities', describes the binding kinetics of these binders, measured by standard procedures
- 'Figure 2. Bio-layer interferometry of nanobody-nup binding' just shows binding isotherms of the different Nanobodies to their different antigens?
- Figure 3 is a high resolution repeat of the right part of figure 1, panel B. More annoying is that the most interesting parts (the ones that are labelled with an Asterix) are the subject of the accompanying paper, proving again that the second manuscript corresponds to the M&M section of the first paper.
- Figure 4 is another example that would be contained in the supplementary figures of a manuscript focussing on the biology of Nup120

- Results section 'Several NPC nanobodies localize to the nuclear envelope *in vivo*' would correspond to the materials and method section on a future high impact paper on cryo-ET studies that makes use of nanobodies to enable subunit identification

In conclusion, this manuscript describes the technical characterization of a number of valuable Nanobodies that bind the NPC. Several of them constitute 'a toolbox' and have great potential to be used for the integrated structural analysis of the Nuclear Pore Complex. Accordingly, the results on two Nanobodies that successfully served this purpose were lifted out of this manuscript and our the subject of the first manuscript.

However, the discussion of the second paper on the remaining Nanobodies is exceedingly hypothetical in its biological interpretation. The short version of this discussion reads like this: '*... suggesting ... suggesting ... may be ... another possibility ... may be ... would suggest ... We hypothesize ... could represent ... or potentially ... most likely... This suggests ... We cannot exclude at present ... might exert ... potentially due ... We hypothesize ... possibly ... it is also possible ... This most likely ... Further investigation ... are hypothesized ... suggests ... likely ... may not be critical ... There are two possible explanations ... Whether ... is also under ongoing investigation*'.

The authors will agree that many of the conclusions obtained on Nup85, Nup120, and Nup133 and the nanobodies binding to these proteins are premature and should be published separately when more data become available. The scientists responsible for the generation and characterization of these valuable research tools should be authors on the first manuscript and included as co-authors on future manuscripts. But with the current data, this feels like two high impact papers for the price of one ...

If however two manuscripts are accepted as accompanying papers, all materials & methods relating to library constructions, phage display and selection should be brought together in the paper focussing on the description of the toolbox. In the current versions, the 'toolbox' paper refers to the 'structure paper' for immunizations, panning, ELISAs, ...?

Minor points:

- The letters C and T (Crown and Tail) are not explained in the legend of figure 1, panel B.
- The interpretation of figure 2 would largely increase if the graphical representations of figure 1, panel B (central part) are reiterated in figure 2.
- Line 187: SAN10 and 11 add to a growing list of nanobodies that bind a variety of differently structured epitopes. It is not entirely clear to me what 'a variety of differently structured epitopes' means? We all agree that the nanobody binds one unique (structural) conformation of the epitope?

Jan Steyaert

VIB-VUB center for Structural Biology

Brussels

REVIEWER COMMENTS

Reviewer #1 (Remarks to the Author):

I recommend publication of this manuscript with minor revisions. See detailed report in attach.

Jan Steyaert

VIB-VUB center for Structural Biology

Brussels

Reviewer #2 (Remarks to the Author):

Dear Thomas et al,

Nordeen et al report the crystal structures of full length γ Nup84 in complex with the C-terminal domain of Nup133 as well as the N-terminal domain of Nup133. The study is the first one to structurally analyse the connection between the Nup107 N and C-terminal domains, the last missing structured piece of the Y-complex, thereby finishing an effort of the field that has been ongoing for ~20 years. The paper further offers the first complete scaffold Nup subcomplex structure from a single species, which is also a milestone because any previous study had to rely to some extent on homology modelling, which comes with some uncertainties. At last, the Nup133 ALPS motif is characterized in yeast, which reveals insights into the conservation of the Nup membrane binding and NPC assembly mechanism.

The study is very well done and an important contribution to NPC structure determination. Overall this is a great paper that I enjoyed reading. I wholeheartedly support its publication in NatComm, after some minor revisions that can be addressed with adjusting the wording.

- Figure 1: I would have found it helpful if there were an inset color-coding/overlaying this with previous structures (highlighting the thus far unknown helical region).

- Do the authors have any idea why is human Nup133 C-term more bent?

- The authors do not really compare their previous homology model to the present full-length structure of Nup84 – and this is also not required for this paper. They however suggest that both composite models can be conceived as two snapshots of the conformational ensemble of the Y-complex, which is a very cool thought. I thus wonder why the previous homology model can be conceived as a possible conformer? I agree with the authors that the nature of the hinges (how rigid they really are) will be an important future research direction. Maybe this could be explained better.

- Lines 280-281: If there is no other way of fitting the Y-complex model into the cryo-EM map, how can it be tentative? Also the respective statement in the discussion is misleading: “However, no cryo-ET map has been solved to high enough resolution to discern secondary structures, which would ultimately allow for unambiguous placement of nups 49.” Ref 49 (Cassidy et al) states on this topic “As the information present at these resolutions is, in general, not sufficient to unambiguously discern a structure’s full topology or, in lower resolution cases, even its tertiary organization, intermediate-resolution approaches typically require the user to provide a complete, atomistic model as input.” These two statements mean something very very different! Cassidy et al emphasize that if secondary structure resolution is not available, independently derived atomic models are a prerequisite for hybrid modelling, which is subsequently explained in length. Cassidy et al by no means state that hybrid modelling yields ambiguous results.

There are certainly limitations of hybrid modelling at intermediate resolution, such as limited precision in spatial assignments of proteins, inability of fitting small structures independently, inability to refine secondary structure elements and side chains, I would even agree to protein interfaces suggested by hybrid modelling remain tentative. But to say that hybrid modelling per se results in ambiguous models, is a massive over-statement. In addition, secondary structure resolution has been recently obtained for the XI NPC (see bioRxiv preprint by Shi lab).

That structural biologists in this field are being conceived as opposing each other harms ourselves the most. In the end, we do agree on many things and we do benefit from each other!

- The discussion may explain better what the benefit of solving an entire subcomplex from a single species really is.

- Line 62: typo ‘do’

Best regards,

Martin

Reviewer #3 (Remarks to the Author):

Nordeen et al describe in their manuscript "Structure of the yeast Nup84-Nup133 complex details flexibility and reveals universal conservation of the membrane anchoring ALPS motif" two nanobody bound structures of *S. cerevisiae* nuclear pore complex (NPC) components: a complex of Nup84 and the C-terminal part of Nup133 and the N-terminal domain of Nup133. Both structures are part of a larger subunit within NPCs, the so called Y-complex. Although structures of the homologues from other species are available, in my eyes this is an important study: high-resolution structural information of all components of the *S. cerevisiae* Y- complex is now available so that a complete composite model of the Y-complex in this important model organism is generated. It shows important differences to the previous homology-based model and can be better fitted in the cryo-EM map of the *S. cerevisiae* NPC. In addition, the structures and the composite models shows important interspecies differences, important to understand the evolutionary aspects of NPC structure, function and assembly.

The manuscript is easy to follow, important for the field and an essential step towards a full high-resolution understanding of NPCs. I have only a few minor points to add:

- 1.) Figure 5: In the liposome floatation assays the Nup133(NTD) Δ ALPS protein could be included. It could serve as appropriate negative control emphasizing the point that the ALPS motif is really crucial for membrane interaction
- 2.) Line 33: "soluble transport" should be replaced by "transport of soluble cargos"
- 3.) Line 96 and 195: "yeast" could be replaced by *S. cerevisiae* given that there exists more than one yeast species
- 4.) line 403: please specify what a "SEN-p protease" is - probably a spelling mistake?

We thank the reviewers for their constructive and fair criticism of our manuscript(s). We have done our best to address the concerns that were raised. Here is our point-by-point response. Response italicized for clarity.

Comments to Reviewer #1:

Line 96. The authors solved a low resolution structure (6.4Å) of the Nup84-Nup133 C-terminal domain in complex with VHH-SAN8 but give no single argument why this is the best they could do?

Line 112. The authors also solved a low resolution structure (7.4Å) of the Nup84-Nup133 C-terminal domain in complex with VHH-SAN8 and VHH-SAN9 but give no single argument why this is the best they could do?

Regarding modest resolution in X-ray crystallography, it is always a matter of speculation what ultimately limits the diffraction quality of a specific crystal. In the text we argue that the intrinsic flexibility of the molecules may well contribute to a crystal of limited order, resulting in limited resolution. As we already argued in the original text, we believe that the nanobodies act as conformation stabilizers. Also, these crystals have very large solvent contents, which typically result in limited resolution as well. We have added text to explain these points better.

Line 162. The authors discuss the phasing of nanobody-free Nup84-Nup133CTD crystals but do not show the data? As these data are used to support an important hypothesis, it would make sense for the experts in the NPC field to describe these results in the supplementary data?

We included the new supplementary figure 6 comparing the density of Nup84-Nup133CTD, VHH-SAN8 bound or nanobody-free. We included collection statistics of the 8.3Å data into Table 1.

Line 184. Does it make sense to talk about an RMSD with an accuracy beyond the comma if you start from 6-7Å resolution structures? On line 258, the RMSD is only given with one figure beyond the comma.

We reduced the RMSDs to include only one decimal place for all comparisons of structures with resolutions worse than 3 Å.

Line 298: What is 'this new model' referring to? 'Two new structures' or 'structural inventory'?

New model was supposed to refer to the new composite model the additional structures allowed us to construct. We adjusted the text to clarify.

Line 746. Please add the resolution of the structure to the legend and indicate the PDB code.

Line 766. Add the PDB code to scup133

Line 772. Add PDB codes

Line 782. Add PDB codes

We use the standard convention of listing resolutions and PDB codes in Table 1 rather than figure legends.

Comments to Reviewer #2:

Figure 1: I would have found it helpful if there were an inset color-coding/overlying this with previous structures (highlighting the thus far unknown helical region).

We expanded Figure 1 to put this structure in context of previously solved Nup84-Nup133 structures.

Do the authors have any idea why is human Nup133 C-term more bent?

We believe this is due primarily to crystal packing, as negative stain EM structures show a very high degree of motion in this particular Nup (see ref. Kim et al, 2018). We added a sentence in the results section citing this previous negative stain EM study.

The authors do not really compare their previous homology model to the present full-length structure of Nup84 – and this is also not required for this paper. They however suggest that both composite models can be conceived as two snapshots of the conformational ensemble of the Y-complex, which is a very cool thought. I thus wonder why the previous homology model can be conceived as a possible conformer? I agree with the authors that the nature of the hinges (how rigid they really are) will be an important future research direction. Maybe this could be explained better.

Figure 6 shows the comparison between our new and our previous model. We added text to better discuss the conformational differences between the two structures/models.

Lines 280-281: If there is no other way of fitting the Y-complex model into the cryo-EM map, how can it be tentative? Also the respective statement in the discussion is misleading: “However, no cryo-ET map has been solved to high enough resolution to discern secondary structures, which would ultimately allow for unambiguous placement of nups 49.” Ref 49 (Cassidy et al) states on this topic “As the information present at these resolutions is, in general, not sufficient to unambiguously discern a structure’s full topology or, in lower resolution cases, even its tertiary organization, intermediate-resolution approaches typically require the user to provide a complete, atomistic model as input.” These two statements mean something very very different! Cassidy et al emphasize that if secondary structure resolution is not available, independently derived atomic models are a prerequisite for hybrid modelling, which is subsequently explained in length. Cassidy et al by no means state that hybrid modelling yields ambiguous results.

There are certainly limitations of hybrid modelling at intermediate resolution, such as limited precision in spatial assignments of proteins, inability of fitting small structures independently, inability to refine secondary structure elements and side chains, I would even agree to protein interfaces suggested by hybrid modelling remain tentative. But to say that hybrid modelling per se results in ambiguous models, is a massive over-statement. In addition, secondary structure resolution has been recently obtained for the XI NPC (see bioRxiv preprint by Shi lab). That structural biologists in this field are being conceived as opposing each other harms ourselves the most. In the end, we do agree on many things and we do benefit from each other!

It was not our intention to pitch structural biologist that are experts in different areas against each other. We hope that it is pretty clear from this manuscript, as well as prior publications from my lab, that we see the approach toward a best structural representation of the NPC as a community effort that takes the dedication of different labs with different expertise. We do believe, however, that we should point out weaknesses and limitations of any technology used. Cell biologist read these papers, and it is up to us to inform the readers appropriately. Just as crystal packing, for example, can be misleading or limiting, is it equally important not to jump to conclusions about the overall structure of the NPC from cryo-ET data, paired with additional constraints, of limited resolution. In our view, this does not diminish in any way the importance of those studies. We have reworded the relevant sections in both manuscripts in the hopes to not offend anyone.

The discussion may explain better what the benefit of solving an entire subcomplex from a single species really is.

We modified the discussion to better explain in which way our experiments move the field forward.

Line 62: typo 'do'

We think 'do' is the proper verb here, but are happy to have the editor take another look.

Comments to Reviewer #3:

1.) Figure 5: In the liposome floatation assays the Nup133(NTD) Δ ALPS protein could be included. It could serve as appropriate negative control emphasizing the point that the ALPS motif is really crucial for membrane interaction

We repeated the floatation assay, now including the Nup133(NTD) Δ ALPS mutant. Using more protein, we could also use Coomassie blue instead of silver for gel staining. The result was as expected, i.e. that the mutant, contrary to the wildtype, did interact poorly with liposomes.

2.) Line 33: "soluble transport" should be replaced by "transport of soluble cargos"

Fixed.

3.) Line 96 and 195: "yeast" could be replaced by *S. cerevisiae* given that there exists more than one yeast species

Fixed.

4.) line 403: please specify what a "SEN-p protease" is - probably a spelling mistake?

*We corrected the name to *bdSEN1*.*

REVIEWERS' COMMENTS

Reviewer #1 (Remarks to the Author):

As the authors conveniently addressed most, if not all remarks of the reviewers, I recommend publication.